# Capture of Fullerenes in Cages and Rings by Forming Metal-π Bond Arene Interactions

**DOI:** 10.3390/ma14123424

**Published:** 2021-06-21

**Authors:** Citlalli Rios, Bertha Molina, Roberto Salcedo

**Affiliations:** 1Instituto de Investigaciones en Materiales, Universidad Nacional Autónoma de México, Circuito Exterior s/n, Ciudad Universitaria, Coyoacán, México City 04510, Mexico; citriogo@yahoo.com.mx; 2Facultad de Ciencias, Universidad Nacional Autónoma de México, Circuito Exterior s/n, Ciudad Universitaria, Coyoacán, México City 04510, Mexico; mlnbrt@ciencias.unam.mx

**Keywords:** trapping of fullerene, molecular cages, theoretical calculations

## Abstract

Nowadays, the task of the selectively capture of fullerene molecules from soot is the subject of several studies. The low solubility of fullerenes represents a drawback when the goal is to purify them and to carry out chemical procedures where they participate. There are different molecules that can act as a kind of cocoon, giving shelter to the fullerene cages in such a way that they can be included in a solution or can be extracted from a mix. In this work, a theoretical study of some known and new proposed organic molecules of this kind is presented. In all cases, the interaction occurs with the help of a metallic atom or ion which plays the role of a bridge, providing a place for a metallocene like interaction to occur. The thermodynamic arguments favoring the formation of this adduct species are addressed as well as the nature of the bond by means QTAIM parameters and frontier molecular orbitals analysis.

## 1. Introduction

Fullerenes have invaded several fields of science and technology. The first report of a carbon nanostructure was C_60_ done by Kroto and his coworkers in 1985 [1] and it represented in those days one of the major surprises in the history of chemistry due to its unique properties, such as high surface to volume ratio, as well as high thermal and chemical stability, they can easily either accept or donate electrons and they show a singular symmetric structure [2]. These features make fullerenes attractive for different applications not only in chemistry but also in engineering, biology, physics and even medicine and its industrial production has been pursued [3,4,5,6,7,8,9]. In this respect, Murayama and coworkers [2] proposed a process to obtain fullerenes in large quantities that can be considered as an industrial product. However, the result is a heterogeneous mass, 80% of soot and 20% of a toluene soluble mixture of C_60_ (60%), C_70_ (25%) and large fullerenes (15%). The poor solubility of fullerenes in different solvents is still a drawback [10,11] and the purification of these species has been performed by instrumental exhaustive methods in lab [12], however there is a strong tendency to find chemical methods to overcome this challenge [13] an example is the functionalization of the cage with soluble chains [14]. Another approach is not to solve fullerene, itself but to transport it inside other larger chemical species to deliver the cages into a determined environment. Those chemical species should be molecular hosts with certain properties: (a) they should be wide enough to support the presence of a fullerene in their cage; (b) they should have some electronic characteristics that allow the interaction with the outer wall of fullerene; and (c) they should be soluble and stable in the chosen solvent.

There is an excellent review in which several metallosupramolecular receptors have been proposed as fullerene hosts [15]. This report recovers information of supramolecular systems that can allocate and release fullerene species based on weak interatomic forces. The encapsulation and immobilization of fullerenes has received much attention because it can enhance their properties to develop specific properties. In this sense, the confinement of fullerenes has led to structures [16] with different features, such as photosensitization [17]. They can be used to tune conductivity of MOFs [18], to develop supramolecular architectures [19] and electric and magnetic nano switches [20]. The present work deals in part with this problem, the long-range interaction of fullerene with different systems carrying electron rich centers is theoretically analyzed. Another approach to this feature would be a strong interaction between fullerene and the molecular trap. In this case, the formation of almost genuine covalent bonds is proposed. However, in this case, the species will not be directly joined, but they will contain an intermediate transition metal atom to form covalent coordinated bonds.

The obvious disadvantage of this kind of compound is the difficult isolation of fullerene which makes a chemical separation necessary. On the other hand, a huge advantage is that fullerene would acquire all the electronic characteristics and will be able to provide them in the new compounds and indeed fullerene would be carried by the MOF into crystals, solvents, etc., giving place to a material with special electronic characteristics.

The molecular traps under study in this contribution have the common characteristic of being large, covalent, organic frameworks [21]. They show a polyhedral or polygonal shape containing aromatic rings which will work as the electronic traps for capturing either the fullerene cages or the intermediate transition metal atom. Some of those molecules have similar analogs which have been already prepared, as will be indicated below, but there is at least one case of a totally local designed species.

## 2. Methods

All calculations were carried out by applying a DFT method based on the combination of Becke’s gradient corrections [22] for exchange and Perdew-Wang’s for correlation [23]. This is the scheme for the B3PW91 method which forms part of the Gaussian16 [24] package. This functional was chosen because it has been demonstrated that it is an excellent option to work with systems in which π bonds of organometallic species are involved [25,26,27]. The calculations were performed using the 6–31G** basis set. In order to confirm that the optimized structures were at a minimum of the potential surfaces, frequency calculations were carried out at the same level of theory. Grimme’s empirical dispersion corrections (G-3) used for evaluating the hydrogen bond energy were carried out using the DFT-D3 method [28] and single point PBE [29] calculations.

Quantum theory of atom-in-molecules analysis [30,31] was executed using Multiwfn 3.7 software [32] to assess the bonding type between the transition metal atoms with the coronene unit on the one hand, and with the fullerene on the other hand.

## 3. Results and Discussion

The three molecular traps are shown in Figure 1.

Molecule **a** is that prepared by Beuerle and his coworkers [33] without changes, whereas molecule **b** is a domestic design of a square framework with coronene fragments in the edges. Molecule **c** is inspired by a macrocycle synthesized by Bain and coworkers [34]. However, the system proposed in this work holds coronene units instead of the porphyrinic ones of the original.

Therefore, there are two ways to trap fullerene cages. Firstly, there is the simple interaction by dispersion forces, mainly π-stacking, which compels the fullerene to make a long-distance interaction with the aromatic fragment of the supramolecular species. Secondly, fullerene is captured in an organometallic complex through a transition metal center and the same aromatic fragment. These two cases will be presented in the next sections.

### 3.1. Π-Stacking

One of the most popular intermolecular interactions that has been studied is the so-called π-stacking [35,36]. This phenomenon is defined as the long-distance electronic communication between the π orbitals of two aromatic species through the space. It is important in supramolecular chemistry [33] because two fragments which are attracted by this force can join to form a more elaborated structure. These interactions can be quantified by computational methods as the consideration of dispersion corrections [37]. The mentioned strategy was adopted in this work.

In all the cases, the dispersion energy results suggest that there is some interaction between fullerene and the molecular trap. The fullerene complexes show 33.9, 35.7, and 36.3 Kcal/mol dispersion energies for molecules **a**, **b****,** and **c****,** respectively. Besides, the nearest distance between a six-membered ring from the fullerene surface and its counterpart in the trap molecule is 5.1 Å on average. This value matches well with the experimental result of 8.09 from an aromatic ring to the geometric center of the sphere of fullerene [38]

Molecule **c** belongs to the O_h_ point group in a static image. Its molecular orbital shapes are shown in Figure 2. It is important to highlight that the LUMO is actually a four-folded degenerated set, whereas the HOMO is a larger five folded degenerated set. This strong electronic accumulation arises from the generation of two sets of accidental degeneration, where e_g_, e_u_, and t_2g_ as irreducible representations come together to form both sets.

The first interesting result is focused on the degenerated set corresponding to the LUMO. Practically all the probability is found on the aromatic rings which lie on the edges, so these regions will accept electrons from other species in search for a place to form a bond.

The HOMO in other context is a set of more diffuse functions. It is found in all the edges with no preference for a special aromatic ring.

The inclusion of fullerene, (see Figure 3) does not cause strong modifications, that is to say the fullerene molecule reaches the inner of the cubic cage without electronic changes in it, nor in the fullerene itself. Therefore, the sets HOMO and LUMO belong to the fullerene alone and the nearest probability function to trap the fullerene is the LUMO + 6, although it is relatively near in terms of energy, with a difference of 4.3 eV from the HOMO.

In the case of the compound **c****,** there is not a defined spatial group. However, all the probabilities in virtual as well as occupied orbitals are centered on the coronene fragments. Therefore, these regions are expected to either accept or donate electrons depending on the species acting as the counterparts of the reaction. This behavior is almost the same in compound **b,** which also bears coronene units as catchers. An important fact is that, in both cases, the molecules show accidental degeneracy on the HOMO. The cause of this situation is not symmetry, but the periodical arrangements which establish similar coronene units with the same energy and behavior in the chain of the molecule. This fact allows to suggest that each molecular trap can capture several fullerene molecules.

Interestingly the presence of fullerene does not contribute to avoid this property. In fact, the complex fullerene-compound **c** (or **b**) shows a molecular orbital diagram with full participation of the HOMO and LUMO combinations of both counterparts, but the important point is that fullerene can also be easily trapped in this jail. The arrangement of the trapped fullerene in the hollow of compound **c** is shown in Figure 4.

### 3.2. Formation of Metallocenes

In this case, there is a third participant in the interaction, a transition metal ion. Therefore, the molecular trap again captures fullerene but not directly. Instead, the capture is achieved through the metal atom which plays the role of a bridge. Indeed, the molecular trap follows a similar process to the one analyzed before, but now the captured species is an exohedral C_60_M.

The capability of fullerene to bond transition metal atoms has been object of study of several groups [39,40,41]. There are examples of six different hapticities [41], although η^5^ [42,43] and η^2^ [44] are the most common cases. The η^6^ case is difficult to find. However, there are some experimental [39] and theoretical [45,46,47,48,49] examples to acknowledge. In the present work, two possibilities are considered: chromium η^6^ and Nickel η^2^ complexes.

Figure 5 shows the shape of the complex arising from the interaction between C_60_Cr fragment and the cubic compound **a**. The corresponding molecular orbital shapes are presented in Figure 6. The most remarkable is that the fragment C_60_M exhibits the behavior of an isolated species because its frontier molecular orbitals are practically transferred to the resultant complex with little change in energy.

The behavior of the C_60_M-molecule **c** is something different because the ring contributes with the HOMO (which is accidentally degenerated and shares a function on each coronene ring in every MO, including the one supporting the fullerene), whereas the fullerene metal complex contributes with the LUMO to produce the electronic interaction. The strong difference in this case is that the trapping is now generated by the formation of a covalent bond (see Figure 7).

The Hirschfield charges analysis shows there are interesting negative values in the fullerene surface. The values are 0.251 in the case of the chromium compound and 0.260 for the nickel one. This fact can be considered together the Wiberg bond order results, in which case the results for the coordination bond between chromium atom and the aromatic ring on the MOF is 1.63, whereas the same value for the bond between the fullerene surface and the chromium atom is 1.58. The same results for the nickel complex are 1.75 and 1.70. Such evidence would suggest there is enough charge transfer to give place to strong covalent coordinated bonds.

Some interesting differences are observed between both C_60_M-molecule **b** compounds induced by the Ni and Cr hapticities, e.g., in the case of η^2^-C_60_Ni, the HOMO (a degenerated set) is localized on the edges of the square where transition metal atoms are placed with their coordination sphere (see Figure 8), while the interaction between fullerene and coronene fragments is found at HOMO-4 (the immediate function after the degenerated set of HOMO). In constrast, in η^6^-C_60_Cr, this interaction is found at HOMO and HOMO-2 (see Figure 9), which are quasi-degenerated too (−5.130 and −5.155 eV respectively). In both cases, LUMO is localized again on the coordination complexes of the edges (Figure 8 and Figure 9). Since the OM analysis confirms the interaction between fullerene and coronene unit via transition metal atoms, the question now is what kind of interaction is this? Looking at the QTAIM parameters (see Table 1), it can be seen that in all cases the electronic density (ρ) and Laplacian ($\nabla^{2}\rho$) at the bond critical point on the fullerene-M and M-coronene bond paths are ρ<0.10 e/bohr^3^ and $\nabla^{2}\rho >0$. Therefore, all bonds correspond to close-shell interactions. Following the classification by Nakanishi et al. [50,51], two categories of close-shell bonding can be identified based on the calculated value for the |V(r)|/G(r) ratio (which reflects the magnitude of the covalent interaction), being V(r) the potential energy density and G(r) the kinetic energy density: if |V(r)|/G(r)<1, no covalence is expected and the so called pure close-shell interaction is obtained (ionic or van der Waals bonding), if |V(r)|/G(r)>1 the interaction is called regular close-shell, and some degree of covalence is expected (e.g., molecular complex through charge transfer). In Table 1, in all cases |V(r)|/G(r)>1, therefore the fullerene-M and M-coronene bonds are regular close-shell interactions, with a small covalence contribution [51,52].

Thus, from the above analysis, we can conclude that these compounds can work as molecular traps of fullerenes. However, its main activity consists in the development of a kind of electrical flux around its perimeter.

An additional feature to note is that the energy gaps of these species (containing the trapped fullerene) are 1.143 and 1.113 eV, respectively. These values allow to classify these substances as good semiconductors or even soft conductors. Therefore, the cages would be useful to trap as well as to generate electronic interesting species.

## 4. Conclusions

The designed supramolecular cages can trap fullerene C_60_ either by π-π stacking interaction or by the formation of coordinated covalent bonds. The cages belong to the MOP family of compounds and have edges that contain strong aromatic centers which constitute the main trap regions because of its large electron clusters. The idea is to achieve the formation of links that can transform weak bonds to real covalent coordinate bonds. The nature of the covalent interaction was studied by means the QTAIM method and it was found that the bonds between the metal atom and fullerene or coronene fragments correspond to a regular close-shell interaction, with a small covalence contribution following the Nakanishi classification. Therefore, the studied species show that real weak covalent bonds and the capture of fullerene units are achieved.

## Figures and Tables

**Figure 1 materials-14-03424-f001:**
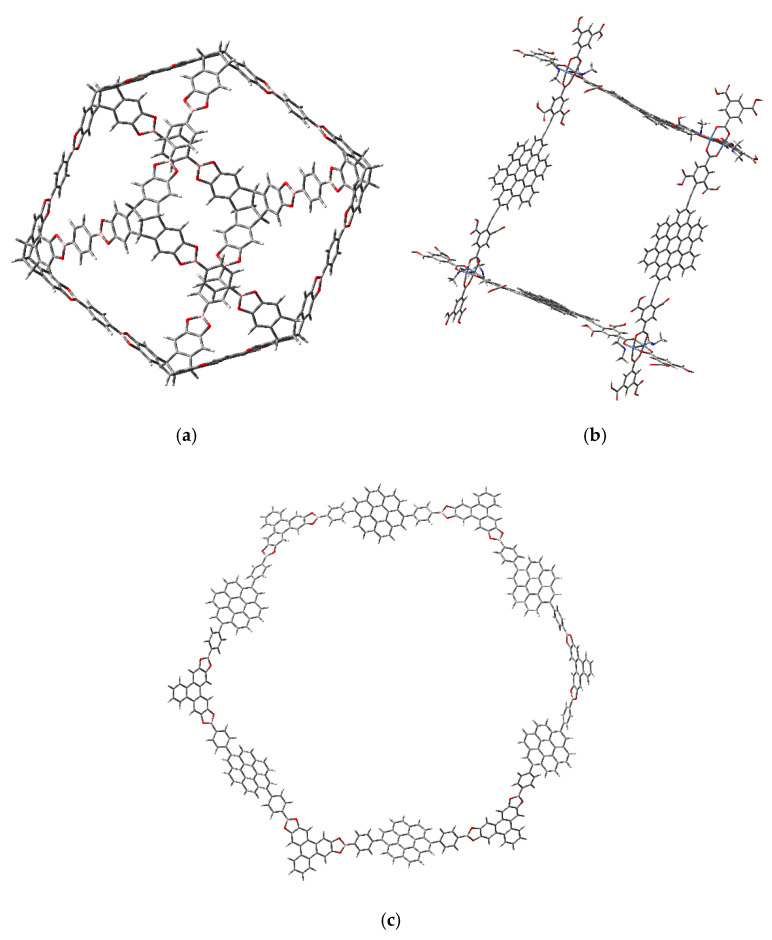
The molecular traps under study. (**a**) A molecule prepared by Beuerle and his coworkers [33]. (**b**) A domestic designed structure with coronene fragments in the edges. (**c**) A molecule inspired by a macrocycle synthesized by Bain and coworkers [34].

**Figure 2 materials-14-03424-f002:**
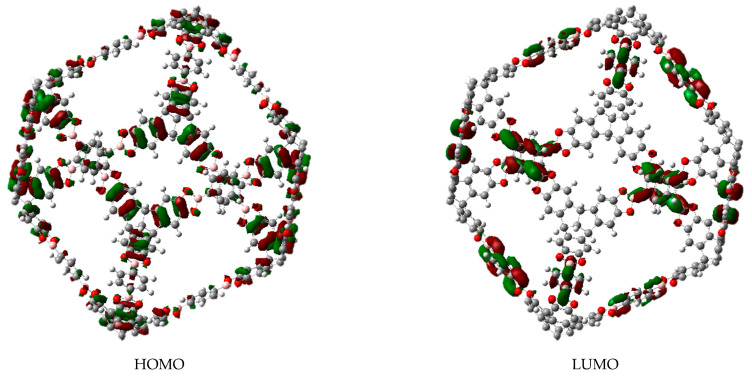
Frontier molecular orbitals of molecule **a**.

**Figure 3 materials-14-03424-f003:**
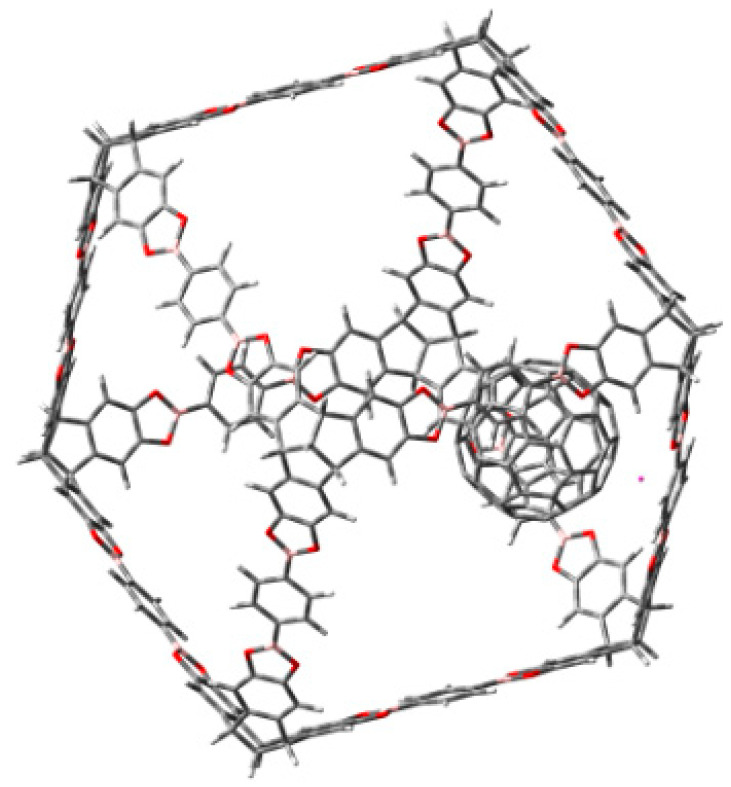
π-stacking interaction between fullerene and molecule **a**.

**Figure 4 materials-14-03424-f004:**
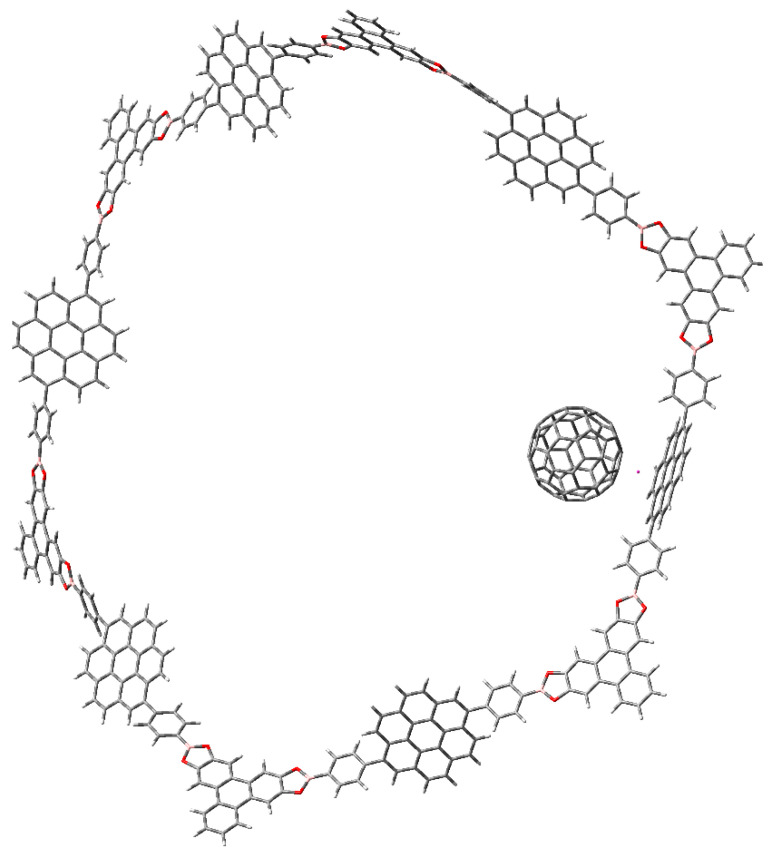
Fullerene molecule trapped into the hollow of molecule **c**.

**Figure 5 materials-14-03424-f005:**
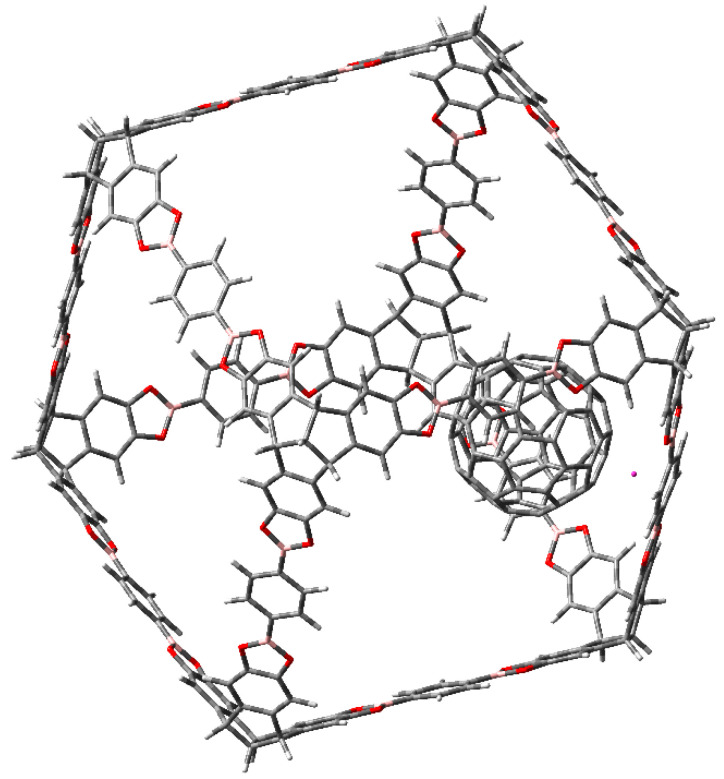
Fullerene and molecule **a** complex with metallocene bond.

**Figure 6 materials-14-03424-f006:**
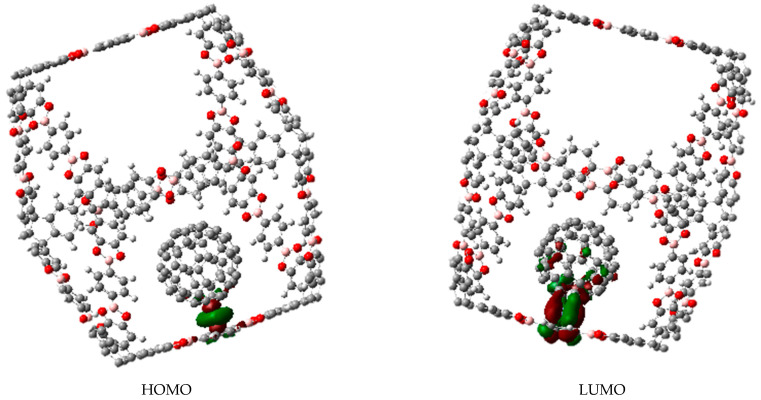
Frontier molecular orbitals of the complex between C_60_M fragment and molecule **a**.

**Figure 7 materials-14-03424-f007:**
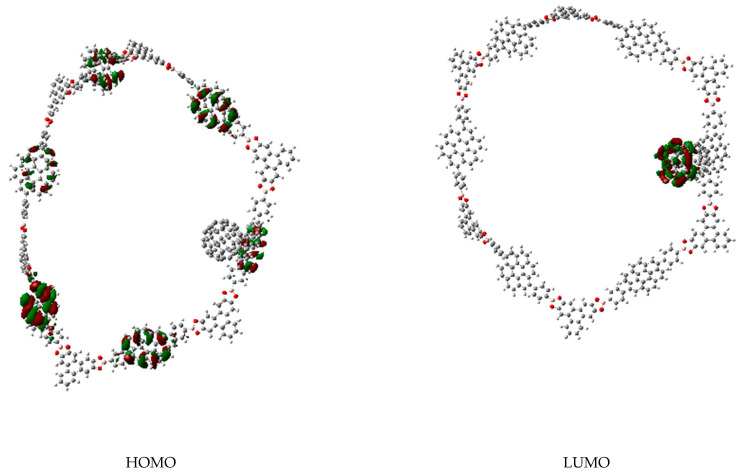
Frontier molecular orbitals of the complex between molecule **c** and C_60_M fragment.

**Figure 8 materials-14-03424-f008:**
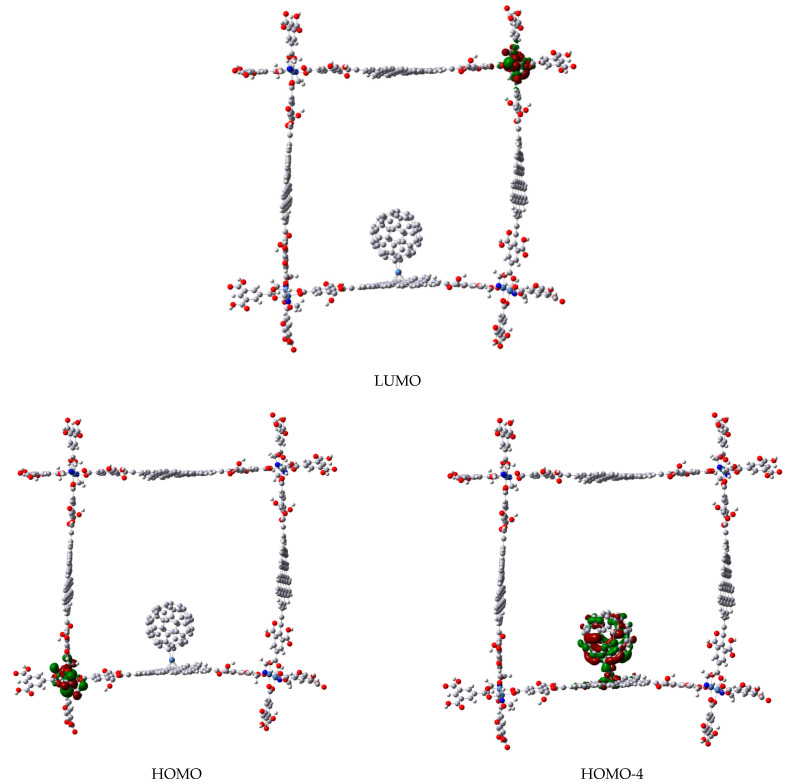
Frontier molecular orbitals of the complex between molecule **b** and C_60_Ni fragment.

**Figure 9 materials-14-03424-f009:**
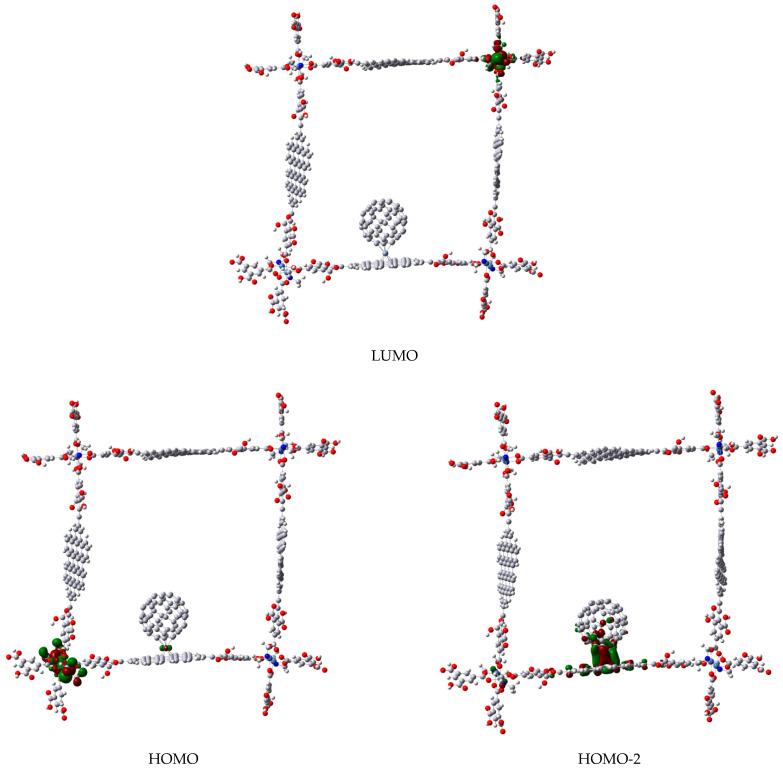
Frontier molecular orbitals of the complex between molecule **b** and C_60_Cr fragment.

**Table 1 materials-14-03424-t001:** QTAIM parameters for C_60_M-Molecule **b** compounds. Bond Critical Point (BCP), electronic density (ρ), Laplacian ($\nabla^{2}\rho$ ), potential energy density (V(r)) and the kinetic energy density (G(r)).

Bond Critical Point Number	ρ u.a.	$\nabla^{2}\rho$ u.a.	G(r) u.a.	|V(r)|/G(r) u.a.
Ni-Coronene	C_60_Ni-Molecule **b**			
BCP1	0.060	0.174	0.064	1.322
BCP2	0.059	0.1780	0.065	1.309
BCP3	0.061	0.179	0.066	1.322
BCP4	0.056	0.190	0.063	1.248
BCP5	0.061	0.168	0.064	1.350
BCP6	0.057	0.188	0.064	1.265
Ni-Fullerene				
BCP1	0.118	0.197	0.114	1.569
BCP2	0.119	0.199	0.115	1.567
Cr-coronene	C_60_Cr-Molecule **b**			
CP1	0.063	0.246	0.072	1.143
CP2	0.059	0.246	0.070	1.119
CP3	0.060	0.243	0.069	1.123
CP4	0.062	0.245	0.071	1.138
CP5	0.062	0.241	0.070	1.142
CP6	0.060	0.246	0.070	1.121
Cr-Fullerene				
CP1	0.054	0.223	0.063	1.114
CP2	0.053	0.216	0.061	1.118
CP3	0.054	0.223	0.063	1.111
CP4	0.052	0.212	0.060	1.118
CP5	0.053	0.220	0.062	1.113
CP6	0.053	0.215	0.061	1.116

## Data Availability

Data sharing not applicable.

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
