# Peer review of "Capture of Fullerenes in Cages and Rings by Forming Metal-π Bond Arene Interactions"

_materials, 2021, doi:10.3390/ma14123424_

Round 1
Reviewer 1 Report
The authors present theoretical studies of new and proposed organic cocoon’s like structures, exploring it interaction with C60 and metallocenes. This encapsulation could improve intrinsic detrimental properties of the fullerene as low solubility. Theoretical studies evaluate the formation of this adduct species and address the nature of the bond by QTAIM parameters and frontier molecular orbitals analysis.
However, some little issues should be improved before it publication.
-Encapsulation of fullerene or others carbon materials within conjugated macrocycles are still being reported and studied in the literature for electronic applications, among other. However, in the article there are not newest citations than 2018. I would suggest an update of the bibliography and a wide explanation of the advantages of encapsulation/inmobilization of fullerenes in molecules. As suggestion, I include two papers related with the work that were published this year and could clarify this comment: Emilio M. Perez et al. Spin-state-dependent electrical conductivity in single-walled carbon nanotubes encapsulating spin-crossover molecules. Nat Commun 12, 1578 (2021) and Ubasart, E., Borodin, O., Fuertes-Espinosa, C. et al. A three-shell supramolecular complex enables the symmetry-mismatched chemo- and regioselective bis-functionalization of C60. Nat. Chem. 13, 420–427 (2021)
-Moreover, the introduction section is not enough wide and broadly explained. The author should expose other techniques/methods that could improve this detrimental properties of the fullerene that they mention. As suggestion, there are some reports about it immobilization of C60 with conjugated polymers using electropolymerization methods that generate two-dimensional films and improve problems of deposition and heterogeneity of the devices.
Author Response
Encapsulation of fullerene or others carbon materials within conjugated macrocycles are still being reported and studied in the literature for electronic applications, among other. However, in the article there are not newest citations than 2018. I would suggest an update of the bibliography and a wide explanation of the advantages of encapsulation/inmobilization of fullerenes in molecules. As suggestion, I include two papers related with the work that were published this year and could clarify this comment: Emilio M. Perez et al. Spin-state-dependent electrical conductivity in single-walled carbon nanotubes encapsulating spin-crossover molecules. Nat Commun 12, 1578 (2021) and Ubasart, E., Borodin, O., Fuertes-Espinosa, C. et al. A three-shell supramolecular complex enables the symmetry-mismatched chemo- and regioselective bis-functionalization of C60. Nat. Chem. 13, 420–427 (2021).
Response:
- The introduction has been modified considering the elements that you pertinently pointed out. Also, this new introduction contains several new and more recent references including those that you kindly suggested.
Moreover, the introduction section is not enough wide and broadly explained. The author should expose other techniques/methods that could improve this detrimental properties of the fullerene that they mention. As suggestion, there are some reports about it immobilization of C60 with conjugated polymers using electropolymerization methods that generate two-dimensional films and improve problems of deposition and heterogeneity of the devices.
Response:
- Besides, the new introduction deals with the problem of obtention and separation of fullerene species from soot as was also suggested.
Reviewer 2 Report
The authors present a computational staudy on capture of fullerenes in three different cages, two of which are designed by them. The work is well planned and conclusion consistent with data reported. However, I hve some reserve on the introduciton, as they report a series of assertions without including some references. The introduction should be improved by proper citations. I support the publication after the following points will be addressed.
Major points
- why did you choose B3PW91, you must explain this choice (literature references or benchmark).
Minor points:
I strongly suggest to replace the optimized structures in Figure 1 with a schematic representations of the traps.
Figures 2-9 should be redesigned by changing the representations of atoms and bonds, maybe tubes representation could be clearer and save them in high resolution. You could also represent fullerene in a different mode, i.e in Figure 5 it is impossible to see the interaction of fullerene with the cage.
1st page
- in the 3rd line of abstract replace where con which.
- please remove the list of properties and separate them with ;
In the conclusion I suggest to add some more sentences to explain the structural characteristic of cages used and the type of fullerene included.
Author Response
The authors present a computational staudy on capture of fullerenes in three different cages, two of which are designed by them. The work is well planned and conclusion consistent with data reported. However, I hve some reserve on the introduciton, as they report a series of assertions without including some references. The introduction should be improved by proper citations. I support the publication after the following points will be addressed.
Response:
- The introduction has been modified and it contains several new and recent references.
why did you choose B3PW91, you must explain this choice (literature references or benchmark).
Response:
- The use of B3PW91 method was justified in the section “Methods”.
I strongly suggest to replace the optimized structures in Figure 1 with a schematic representations of the traps.
Figures 2-9 should be redesigned by changing the representations of atoms and bonds, maybe tubes representation could be clearer and save them in high resolution. You could also represent fullerene in a different mode, i.e in Figure 5 it is impossible to see the interaction of fullerene with the cage.
Response:
- We have changed all the figures which does not contain molecular orbitals surfaces, the new figures have tubular drawing and are clearer.
1st page
- in the 3rd line of abstract replace where con which.
- please remove the list of properties and separate them with ;
Response:
- The typos were corrected.
In the conclusion I suggest to add some more sentences to explain the structural characteristic of cages used and the type of fullerene included.
Response:
- We have a new and wide section of Conclusions considering your suggestions.
Reviewer 3 Report
Specific comments:
- In order to ensure the reliability of the findings, the results of frequency calculations need to be provided.
- Did the authors analyze the charge transfer between C60 and the studied molecules?
- There is a lack of recent references.
Author Response
In order to ensure the reliability of the findings, the results of frequency calculations need to be provided.
Response:
- We have included the output of one of our frequencies calculations as supplementary material.
Did the authors analyze the charge transfer between C60 and the studied molecules?
Response:
- We have carried out an analysis of Hirschfiel charges and Wiberg indexes to assess the charge transfer between fullerene cage and metal atoms and between the aromatic center of the MOP and metal atoms.
There is a lack of recent references.
Response:
- The introduction has been modified and it contains several new and recent references.
Round 2
Reviewer 3 Report
The authors have revised the manuscript carefully according to the reviewer's suggestions.